# Role of TLR4 signaling on *Porphyromonas gingivalis* LPS-induced cardiac dysfunction in mice

**Ichiro Matsuo[1,2], Naoya Kawamura[1,2], Yoshiki Ohnuki[1], Kenji Suita[1], Misao Ishikawa[3], Takehiro Matsubara[4], Yasumasa Mototani[1], Aiko Ito[5], Yoshio Hayakawa[1,6], Megumi Nariyama[7], Akinaka Morii[1,2], Kenichi Kiyomoto[1,2], Michinori Tsunoda[1,2], Kazuhiro Gomi[2], Satoshi Okumura[1]***

1 Department of Physiology, Tsurumi University School of Dental Medicine, Yokohama, Japan,
2 Department of Periodontology, Tsurumi University School of Dental Medicine, Yokohama, Japan,
3 Department of Oral Anatomy, Tsurumi University School of Dental Medicine, Yokohama, Japan, 4 Division of BioBank, Center for Comprehensive Genomic Medicine, Okayama University Hospital, Okayama, Japan,
5 Department of Orthodontics, Tsurumi University School of Dental Medicine, Yokohama, Japan,
6 Department of Dental Anesthesiology, Tsurumi University School of Dental Medicine, Yokohama, Japan,
7 Department of Pediatric Dentistry, Tsurumi University School of Dental Medicine, Yokohama, Japan

* okumura-s@tsurumi-u.ac.jp

**Data Availability Statement:** All relevant data are within the paper and its Supporting Information files.

## Abstract

Oral infections, particularly periodontitis, are a well-established risk factor for cardiovascular diseases, although the molecular mechanisms involved remain elusive. The aims of the present study were to investigate the effects of lipopolysaccharide derived from *Porphyromonas gingivalis* (PG-LPS) on cardiac function in mice, and to elucidate the underlying mechanisms. Mice (C57BL/6) were injected with PG-LPS (0.8 mg/kg/day) with or without an inhibitor of Toll-like receptor 4 (TLR4) signaling (TAK-242, 0.8 mg/kg/day) for 4 weeks. Left ventricular ejection function was significantly decreased at 1 week (from 67 ± 0.5 to 58 ± 1.2%) and remained low at 4 weeks (57 ± 1.0%). The number of apoptotic myocytes was increased (approximately 7.4-fold), the area of fibrosis was increased (approximately 3.3-fold) and the number of 8-hydroxydeoxyguanosine-positive myocytes, a sensitive indicator of oxidative DNA damage, was increased (approximately 7.6-fold) at 4 weeks in the heart of PG-LPS treated mice. However, levels of various serum pro-inflammatory cytokines in PG-LPS-treated mice were similar to those in control mice. The impairment of cardiac function in PG-LPS-treated mice appears to involve activation of TLR4-NADPH oxidase (NOX) 4 signaling, leading to abundant production of reactive oxygen species and $Ca^{2+}$ leakage from sarcoplastic reticulumn induced by calmodulin kinase II (CaMKII)-mediated phosphorylation of phospholamban (at Thr-17) and ryanodine receptor 2 (at Ser-2448). Pharmacological inhibition of TLR4 with TAK-242 attenuated the changes in cardiac function in PG-LPS-treated mice. Our results indicate that TLR4-NOX4 signaling may be a new therapeutic target for treatment of cardiovascular diseases in patients with periodontitis.

**Funding:** This study was supported by the Japan Society for the Promotion of Science (JSPS) KAKENHI Grant (20K10304 to YO, 20K10305 to KS, 21K17171, 19K24109 to AI, 17K11977 to MN, and 21K10242, 18K06862 to SO); Research Promotion Grant from the Society for Tsurumi University School of Dental Medicine (IM and NK). The founders had no role in study design, data collection and analysis, decision to publish, or preparation of the manuscript.

**Competing interests:** The authors declared that no competing interests exist.

## Introduction

Periodontitis (PD) is a bacterial infection in the tissue that supports the teeth, leading to chronic local inflammation, destruction of connective tissue and alveolar bone, and eventually loss of teeth. It is also associated with an increase of myocardial cell damage and heart failure [1,2]. *Porphyromonas gingivalis* (PG) is a gram-negative bacterial pathogen often identified in adult PD, and systemic exposure to pro-inflammatory factors from PG, including lipopolysaccharide, may contribute to CVD [3]. Lipopolysaccharide (LPS), a cell wall component of gram-negative bacteria, plays a major role in the inflammatory response following bacterial infection [4], as well as in the myocardial depression seen in adult patients with sepsis or severe gram-negative bacterial infection [5]. But, even though PG-LPS is widely accepted as a contributor to PD-induced systemic inflammation, and patients with high anti-PG antibody levels were reported to exhibit a high rate of heart failure compared to the low antibody group [6], little is known about the precise relationship between PG-LPS and cardiac dysfunction.

Although PG-LPS and *Escherichia coli* LPS are both derived from gram-negative bacteria, they differ in structure and function. The two types of LPS differentially modulate expression of Toll-like receptor (TLR) 2, TLR4, and cell surface receptor and differentiation marker (CD) 14, as well as primary and secondary cytokine responses [7]. *Escherichia coli* LPS targets TLR4 and activates the NF-κB signaling pathway, leading to the secretion of inflammatory cytokines, such as tumor necrosis factor α (TNF-α) and interleukin 6 (IL-6), and chemokines, such as monocyte chemoattractant protein-1 (MCP-1) [8]. The receptor of PG-LPS was initially reported to be TLR2, in contrast with the well-established role of TLR4 as the receptor for *Escherichia coli* LPS [9]. However, this proved controversial, because synthetic PG lipid A activates TLR4 but not TLR2 [10,11]. More recently, PG-LPS was shown to mediate pro-inflammatory signaling exclusively through TLR4, while activation of the TLR2-dependent pathway was related to be the presence of contaminants in the LPS preparation [8].

Persistently elevated LPS levels are found in chronic diseases, including PD, and thus cardiac cells expressing TLR4 may be a target for PG-LPS [12]. Activation of the innate immune system in the heart by TLR4 has diverse effects, which may be cardioprotective in the short-term, whereas sustained activation may be maladaptive [13]. More importantly, to our knowledge there has been no mechanistic study of the myocardial effects of persistent subclinical exposure to PG-LPS, even though subclinical endotoxemia was previously reported to induce myocardial cell damage and heart failure [14].

Therefore, the aims of this study were to evaluate cardiac dysfunction in mice treated with PG-LPS at a dose equivalent to the circulating levels in PD patients to examine the effects of a TLR4 antagonist (6R)-6-[N-(2-chloro-4-fluorophenyl)sulfamoyl] cyclohex-1-ene-1-carboxylate (TAK-242) [15,16] on the changes of cardiac function, and to clarify the mechanisms involved.

## Materials and methods

### Mice and experimental protocols

All experiments were performed on male 12-week-old C57BL/6 mice obtained from CLEA Japan (Tokyo, Japan). Mice were group-housed at 23˚C under a 12–12 light/dark cycle with lights on at 8:00 AM in accordance with standard conditions for mouse studies by our group [17–20]. Both food and water were available ad libitum.

PG-LPS (Wako, Osaka, Japan) was dissolved in saline to prepare a 0.6 mg/ml stock solution [15] and TAK-242 (ChemScene, Monmouth Junction, NJ, USA) was formulated with 1% dimethyl sulfate and double-distilled water to prepare a 0.4 mg/ml stock solution. The

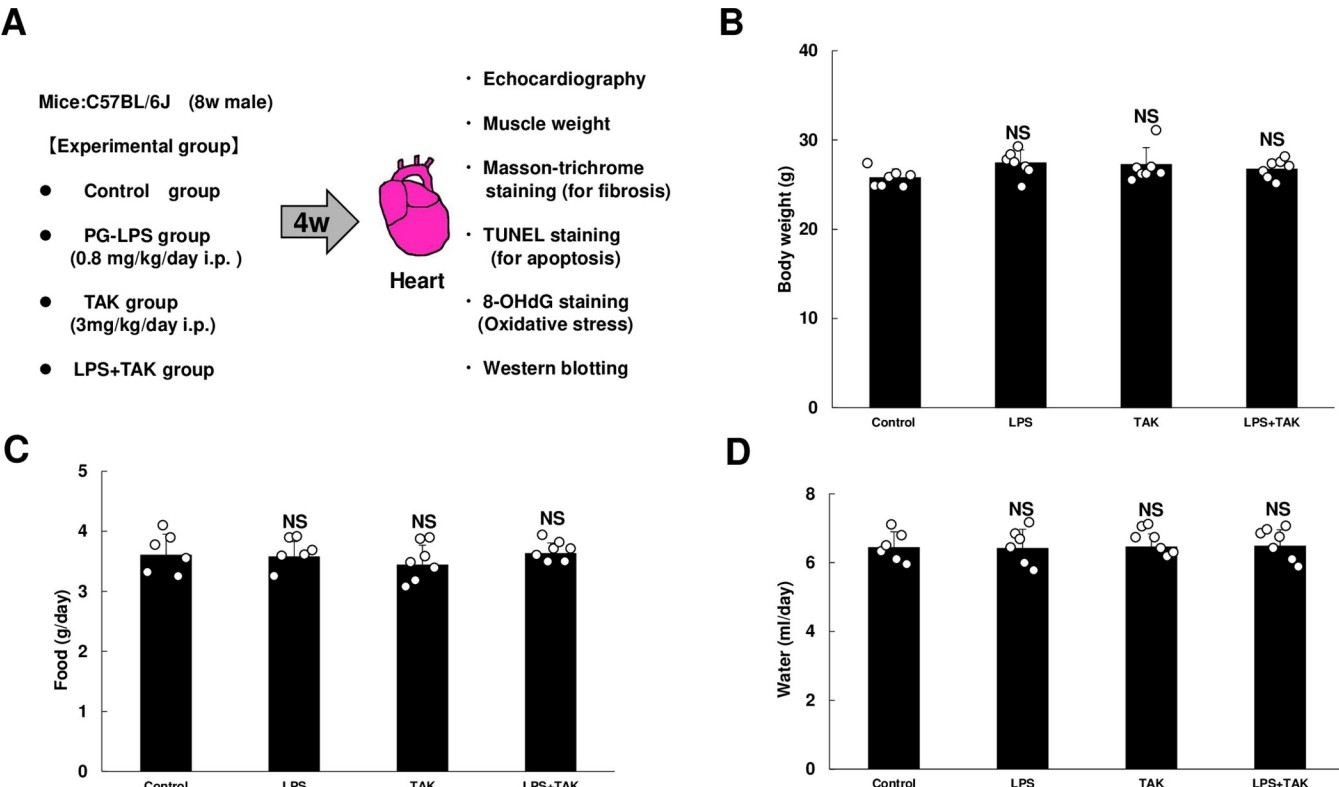

**Fig 1. Experimental procedure and consumption of food and water during chronic PG-LPS infusion in mice. (A)** Lipopolysaccharide derived from *Porphylomonas gingivalis* (PG-LPS) was administered once daily for 4 weeks via intraperitoneal injection (i.p.) at a dose of 0.8 mg/kg, dissolved in saline. Age-matched control mice (Control) received an identical volume of saline only. **(B)** The Control, PG-LPS, TAK and LPS + TAK groups showed similar body weight at 4 weeks after the PG-LPS infusion. NS, not significantly different from the Control ($P > 0.05$) by one-way ANOVA followed by the Tukey-Kramer *post hoc* test. **(C-D)** Consumed amounts of food **(C)** and water **(D)** were similar among the four groups. $P$ = NS vs. Control). NS, not significantly different from the Control ($P > 0.05$) by one-way ANOVA followed by the Tukey-Kramer *post hoc* test. Data shows means ± SD and scattered dots show individual data.

appropriate volumes of these solutions to provide the desired dose (PG-LPS: 0.8 mg/kg and/or TAK-242: 3 mg/kg) was added to 0.2 ml of saline to prepare the solution for intraperitoneal (i. p.) injection (once daily for 4 weeks); control mice received an identical volume of saline only [21] **(Fig 1A)**. In addition, BW and intake of food and water were monitored for all mice throughout the 4-week experimental period. The dose of PG-LPS used in this study is consistent with the circulating levels in PD patients, indicating that this model is not a sepsis model, and no mortality was observed [15]. After the completion of treatment, mice were anesthetized with isoflurane. The heart, lungs and liver were excised, weighed, frozen in liquid nitrogen, and stored at -80˚C. The ratio of organ mass (mg) to tibia length (TL; mm) were used as indexes of organ volume. After tissue extraction, the mice were anesthetized via a mask with isoflurane (1.0–1.5% v/v) and killed by cervical dislocation [22].

## Ethical approval

All animal experiments complied with the ARRIVE guidelines [23] and were carried out in accordance with the National Institutes of Health guide for the care and use of laboratory animals [24] and institutional guidelines. The experimental protocol was approved by the Animal Care and Use Committee of Tsurumi University (No. 29A041).

## Physiological experiments

Mice were anesthetized with isoflurane vapor (1.0–1.5% v/v) titrated to maintain the lightest anesthesia possible and echocardiographic measurements were performed by using ultrasonography (TUS-A300, Toshiba, Tokyo, Japan) as described previously [25].

## Evaluation of fibrosis

Among several quantitative methods available to determine interstitial fibrotic regions [18,26,27], we employed Masson-trichrome staining using the Accustatin Trichrome Statin Kit (#HT15-1KT; Sigma-Aldrich, St. Louis, MO, USA) in accordance with the manufacturer's protocol, as described previously. We quantified interstitial fibrotic regions using freely available image analysis software (Image J 1.45; https://imagej.nih.gov/ij/download.html) to evaluate the percentage of blue area in the Masson-trichrome sections [18].

## Evaluation of apoptosis

Apoptosis was determined by TUNEL staining using an Apoptosis *in situ* Detection Kit (#293–71501; Wako, Osaka, Japan). TUNEL-positive nuclei per field of view were manually counted in six sections of each of the four groups (Control, PG-LPS, TAK-242, PG-LPS + TAK-242) over a microscopic field of 20 x, averaged and expressed as the ratio of TUNEL-positive nuclei (%) [25,28]. Limiting the counting of total nuclei and TUNEL-positive nuclei to areas with true cross sections of myocytes made it possible to selectively count only those nuclei that were clearly located within myocytes.

## Bio-Plex measurement of pro-inflammatory cytokines, PDGF-BB and VEGF

Serum levels of pro-inflammatory cytokines IL-6, IL-1β, IL-10, IL-17, IFN-γ, MCP-1 and TNF-α, as well as biomarkers of endothelial function, PDGF-BB and VEGF, were measured using a multiplex suspension array (Bio-Plex, Bio-Rad, Hercules, CA, USA) [29].

## Western blotting

The cardiac muscle excised from the mice (**Fig 1A**) was homogenized in a Polytron (Kinematica AG, Lucerne, Switzerland) in ice-cold RIPA buffer (Thermo Fisher Scientific, Waltham, MA, USA: 25 mM Tris-HCl (pH 7.6), 150mM NaCl, 1% NP-40, 1% sodium deoxycholate, 0.1% SDS) without addition of inhibitors [30], and the homogenate was centrifuged at 13,000 x *g* for 10 min at 4˚C. The supernatant was collected and the protein concentration was measured using a DC protein assay kit (Bio-Rad, Hercules, CA, USA). Equal amounts of protein (5 μg) were subjected to 12.5% SDS-polyacrylamide gel electrophoresis and blotted onto 0.2 mm PVDF membrane (Millipore, Billerica, MA, USA).

Western blotting was conducted with commercially available antibodies [25,28,31]. The primary antibodies against TLR4 (1:1000, #14358), α-smooth muscle actin (α-SMA) (1:1000, 19245), Akt (1:1000, #9272), phospho-Akt (Ser-473, 1:1000, #9721), CaMKII (1:1000, #3362), phospho-CaMKII (1:1000, Thr-286, #3361), ERK (1:1000, #4695), phospho-ERK (1:2000, Thr-202/Tyr-204, #4370), BCL-2 (1:1000, #3498), LC3 (1:1000, #12741) and phospho- mTOR (1:1000, Ser-2448, #5536), mTOR (1:1000, #2972), p70S6K (1:1000, #9202), phospho-p70S6K (1:1000, Thr-389), RIP3 (1:1000, #95702), phospho-RIP3 (1:1000, Thr231/Ser232, #91702), AMPK (1:1000, #2532) and phospho-AMPK (1:1000, Thr-172, #2535) were purchased from Cell Signaling Technology (Boston, MA, USA), anti-ox-CaMKII was purchased from Millipore (1:1000, #07–1387, Billerica, MA, USA), the primary antibodies against glyceraldehyde-

3-phosphate dehydrogemase (GAPDH) (1:200, sc-25778) were purchased from Santa Cruz Biotechnology (Santa Cruz, CA, USA) and the primary antibodies against phospho-PLN (1:5000, Thr-17, #A010-13), PLN (1:5000, #A010-14), phospho- RyR2 (1:2000, Ser-2808, #A010-30) and phospho-RyR2 (Ser-2814, #A010-31) were purchased from Badrilla (Leeds, UK), while RyR2 (1:1000, #MA3-916) was purchased from Thermo Fisher (Rockland, IL, USA). The primary antibodies against p62 (1:1000, #PM045) and phospho-p62 (1:500, Ser-351) (#PM074MS) were purchased from MBL (Nagoya, Japan) and the primary antibodies against NOX4, 1:1000, #ab133303) and NOX2 (1:1000, #ab80508) were purchased from Abcam (Cambridge, UK). Horseradish peroxide-conjugated anti-rabbit (1:5000, #NA934) or anti-mouse IgG (1:5000, #NA931) purchased from GB Healthcare was used as a secondary antibody. The primary and secondary antibodies were diluted in Tris-buffered saline (pH 7.6) with 0.1% Tween 20 and 5% bovine serum albumin. The blots were visualized with enhanced chemiluminescence solution (ECL:Prime Western Blotting Detection Reagent, GE Healthcare, Piscataway, NJ, USA) and scanned with a densitometer (LAS-1000, Fuji Photo Film, Tokyo, Japan).

## Immunostaining

Oxidative DNA damage in the myocardium was evaluated by immunostaining for 8-OHdG using the Vector M.O.M Immunodetection system (#PK-2200, Vector Laboratories, Inc. Burlingame, CA, USA) [19,20,32,33]. Cross sections were cut with a cryostat at -20˚C at 10 μm, air-dried and fixed with 4% paraformaldehyde (v/v) in TBS-T for 5 min at room temperature. Antigen retrieval was achieved with 0.1% citrate plus 1% Triton X-100 for 30 min at room temperature, then the sections were washed with TBS-T, incubated with 0.3% horse serum in TBS-T for 1 h at room temperature, and blocked with M.O.M. blocking reagents (Vector Laboratories, Burlingame, CA, USA) overnight at 4˚C. For the positive control, sections were incubated with 0.3% $H_2O_2$ in TBS-T before the anti-8-OHdG antibody treatment. The sections were incubated with anti-8-OHdG antibody (8.3 μg/ml in M.O.M. Dilute; clone N45.1 monoclonal antibody; Japan Institute for the Control of Aging, Shizuoka, Japan) overnight at 4˚C in a humidified chamber, and then incubated with 0.3% $H_2O_2$ in 0.3% horse serum for 1 h at room temperature to inactivate endogenous peroxidase, rinsed with TBS-T, incubated with anti-mouse IgG in M.O.M. Diluent, and processed with an ABC kit (Vector Laboratories, Inc. Burlingame, CA, USA). We calculated the ratio of 8-OHdG nuclei with oxidative DNA damage (stained dark blown) per total cell numbers.

## Statistical analysis

Data show means ± standard deviation (SD). Comparison of data was performed using one-way ANOVA followed by Tukey's *post hoc* test. Differences were considered significant when $P < 0.05$.

## Results

### Body weight, daily consumption of food and water

The Control, PG-LPS, TAK-242, PG-LPS + TAK-242 groups all showed similar body weight (BW) at 4 weeks after the PG-LPS infusion (PG-LPS [$n = 7$]: 27.4 ± 0.6, TAK-242 [$n = 7$]: 27.2 ± 0.7, PG-LPS + TAK-242 [$n = 7$]: 26.7 ± 0.4 g, all not significantly different [NS; $P > 0.05$] vs. Control [$n = 7$; 25.8 ± 0.4 g]) (Fig 1B), and the consumed amounts of both food (Fig 1C) and water (Fig 1D) were also similar among the four groups (food: PG-LPS [$n = 7$]: 3.6 ± 0.1, TAK [$n = 7$]: 3.4 ± 0.1, PG-LPS + TAK-242 [$n = 7$]: 3.6 ± 0.1 g, all NS [$P > 0.05$] vs.

Control [3.6 ± 0.1 g]; water: PG-LPS [$n$ = 6]: 6.4 ± 0.2, TAK-242 [$n$ = 7]: 6.5 ± 0.1, PG-LPS + TAK-242 [$n$ = 7]: 6.5 ± 0.2 ml/day, all NS [$P$ > 0.05] vs. Control [$n$ = 6; 6.4 ± 0.2 ml/day] each).

Thus, chronic PG-LPS treatment with or without TAK-242 under the experimental conditions used in this study did not appear to affect growth, or food and water consumption.

## Effects of PG-LPS on heart, lung and liver weight

Mice in the four groups were sacrificed at 4 weeks, and the cardiac muscle (CA) was weighed to evaluate cardiac hypertrophy, and the lungs and liver were weighed to estimate the pulmonary and liver congestion.

Heart weight in terms of the ratio of CA weight per tibia length (TL) was not significantly different among the four groups (CA/TL: PG-LPS [$n$ = 7]: 7.5 ± 0.3, TAK [$n$ = 7]: 7.2 ± 0.5, PG-LPS + TAK-242 [$n$ = 7]: 7.7 ± 0.3, all NS [$P$ > 0.05] vs. Control [$n$ = 7; 7.5 ± 0.4 g] each) (S1 Fig of S1 Data). The lung weight in terms of the ratio of lung weight per TL (S1B Fig of S1 Data), and liver weight in terms of the ratio of liver weight per TL (S1C Fig of S1 Data) were also similar among the four groups (lung weight/TL: PG-LPS [$n$ = 7]: 8.9 ± 0.4, TAK [$n$ = 7]: 9.2 ± 0.3, PG-LPS + TAK-242 [$n$ = 7]: 9.0 ± 0.2 mg/mm, all NS [$P$ > 0.05] vs. Control [$n$ = 7; 8.6 ± 0.3 mg/mm]; liver weight/TL: PG-LPS [$n$ = 7]: 85.1 ± 3.1, TAK [$n$ = 7]: 73.1 ± 2.9, PG-LPS + TAK-242 [$n$ = 7]: 73.2 ± 3.6 mg/mm, all NS [$P$ > 0.05] vs. Control [$n$ = 7; 80.3 ± 2.9 mg/mm]).

Thus, neither PG-LPS nor TAK-242 at the dose used in this experiment appeared to influence the weight of the heart, liver or lungs during the 4-week experimental period.

## Serum levels of pro-inflammatory cytokines, PDGF-BB and VEGF

We examined the effects of chronic PG-LPS infusion on serum levels of IL-6, IL-1β, IL-10, IL-17, interferon-γ (IFN-γ), MCP-1, TNF-α, as well as platelet-derived growth factor-BB (PDGF-BB) and vascular endothelial growth factor (VEGF). The serum levels of these molecules were similar among the four groups (Table 1, S1-S5 Figs of S2 Data), indicating that chronic PG-LPS infusion had no effect on them.

## Changes of cardiac function during the chronic PG-LPS infusion

Echocardiography was performed at 1 week (Table 2A) and 4 weeks (Table 2B) after the start of chronic PG-LPS treatment with or without TAK-242.

**Table 1. Serum levels of pro-inflammatory cytokines, PDGF-BB and VEGF.**

| pg/ml | Control | LPS | TAK | LPS+TAK |
|---|---|---|---|---|
| n | 7 | 7 | 7 | 7 |
| TNF-α | 1343 ± 361 | 1079 ± 300 | 1335 ± 698 | 772 ± 400 |
| IL-1β | 44 ± 11 | 43 ± 8 | 51 ± 18 | 45 ± 20 |
| IL-6 | 26 ± 6 | 22 ± 6 | 27 ± 5 | 21 ± 9 |
| IL-10 | 170 ± 33 | 146 ± 12 | 192 ± 22 | 146 ± 45 |
| IL-17 | 243 ± 144 | 250 ± 54 | 187 ± 77 | 161 ± 103 |
| IFN-γ | 152 ± 33 | 143 ± 26 | 243 ± 210 | 128 ± 44 |
| MCP-1 | 415 ± 87 | 420 ± 56 | 420 ± 94 | 410 ± 116 |
| VEGF-A | 86 ± 10 | 85 ± 9 | 95 ± 15 | 85 ± 26 |
| PDGF-BB | 8648 ± 6188 | 9476 ± 7982 | 7491 ± 5604 | 10238 ± 9420 |

**Table 2. Cardiac function assessed by echocardiography.**

**A. One week after PG-LPS**

| | Control | LPS | TAK | LPS+TAK |
|---|---|---|---|---|
| **n** | 7 | 7 | 7 | 7 |
| **EF** | 67 ± 0.5 | 58 ± 1.2[**] | 62 ± 0.7 | 60 ± 0.8 |
| **LVEDV** | 0.20 ± 0.01 | 0.21 ± 0.01 | 0.21 ± 0.009 | 0.23 ± 0.01 |
| **LVESV** | 0.066 ± 0.005 | 0.090 ± 0.005 | 0.078 ± 0.003 | 0.091 ± 0.006 |
| **FS** | 32 ± 0.3 | 26 ± 0.7[**] | 29 ± 0.5 | 27 ± 0.5 |
| **LVDd** | 4.3 ± 0.09 | 4.4 ± 0.09 | 4.4 ± 0.07 | 4.5 ± 0.08 |
| **LVDs** | 2.9 ± 0.07 | 3.3 ± 0.06[**] | 3.1 ± 0.05 | 3.3 ± 0.08 |
| **HR** | 368 ± 27.6 | 366 ± 19.6 | 355 ± 17.3 | 337 ± 13.0 |
| **SV** | 0.13 ± 0.008 | 0.12 ± 0.009 | 0.13 ± 0.006 | 0.14 ± 0.006 |
| **CO** | 50 ± 5.2 | 45 ± 4.5 | 46 ± 3.1 | 46 ± 3.0 |
| **IVSDd** | 0.55 ± 0.02 | 0.55 ± 0.02 | 0.52 ± 0.01 | 0.56 ± 0.017 |
| **IVSDs** | 0.96 ± 0.03 | 0.89 ± 0.02 | 0.89 ± 0.03 | 0.90 ± 0.03 |
| **PWTd** | 0.53 ± 0.02 | 0.53 ± 0.002 | 0.53 ± 0.019 | 0.54 ± 0.011 |
| **PWTs** | 0.83 ± 0.02 | 0.78 ± 0.015 | 0.83 ± 0.02 | 0.82 ± 0.02 |

**B. Four weeks after PG-LPS**

| | Control | LPS | TAK | LPS+TAK |
|---|---|---|---|---|
| **EF** | 66 ± 0.5 | 57 ± 1.0[**] | 60 ± 1.3 | 61 ± 0.6[#] |
| **LVEDV** | 0.22 ± 0.01 | 0.23 ± 0.02 | 0.24 ± 0.017 | 0.24 ± 0.02 |
| **LVESV** | 0.073 ± 0.003 | 0.097 ± 0.007 | 0.096 ± 0.010 | 0.092 ± 0.006 |
| **FS** | 32 ± 0.3 | 26 ± 0.7[**] | 27 ± 0.8 | 28 ± 0.4[#] |
| **LVDd** | 4.4 ± 0.06 | 4.5 ± 0.11 | 4.6 ± 0.11 | 4.6 ± 0.11 |
| **LVDs** | 3.0 ± 0.05 | 3.3 ± 0.09 | 3.3 ± 0.12 | 3.3 ± 0.07 |
| **HR** | 409 ± 15.0 | 412 ± 22.7 | 377 ± 24.4 | 396 ± 18.4 |
| **SV** | 0.14 ± 0.005 | 0.13 ± 0.010 | 0.14 ± 0.008 | 0.14 ± 0.01 |
| **CO** | 59 ± 4.3 | 53 ± 4.3 | 54 ± 5.0 | 57 ± 3.8 |
| **IVSTd** | 0.50 ± 0.02 | 0.49 ± 0.02 | 0.51 ± 0.02 | 0.51 ± 0.017 |
| **IVSTs** | 0.91 ± 0.02 | 0.81 ± 0.03 | 0.84 ± 0.04 | 0.85 ± 0.02 |
| **PWTd** | 0.49 ± 0.03 | 0.48 ± 0.01 | 0.50 ± 0.013 | 0.53 ± 0.018 |
| **PWTs** | 0.86 ± 0.03 | 0.80 ± 0.03 | 0.84 ± 0.02 | 0.86 ± 0.03 |

EF (%): Ejection fraction.

LVEDV (mL): Left ventricular end-diastolic volume.

LVESV (mL): Left ventricular end-systolic volume.

FS (%): Fractional shortening.

LVDd (mm): Left ventricular internal dimension at end-diastole.

LVDs (mm): Left ventricular internal dimension at end-systole.

HR (bpm): Heart rate.

SV (mL): Stroke volume.

CO (mL/min): Cardiac output.

IVSTd (mm): Interventricular septum thickness at end-diastole.

IVSTd (mm): Interventricular septum thickness at end-systole.

PWTd (mm): Posterior wall thickness at end-diastole.

PWTs (mm): Posterior wall thickness at end-systole.

[**]$P < 0.01$ vs. Control.

[#]$P < 0.05$ vs. LPS.

Cardiac function in terms of left ventricular ejection fraction (EF) and %fractional shortening (FS) was significantly decreased at 1 week after chronic PG-LPS injection (EF: Control [$n = 7$] vs. PG-LPS [$n = 7$]: 67 ± 0.5 vs. 58 ± 1.2%, $P < 0.01$; FS: Control [$n = 7$] vs. PG-LPS

[$n$ = 7]: 32 ± 0.3 vs. 26 ± 0.7%, $P$ < 0.01, **S6A and S7B Figs of S2 Data**) with an increase of left ventricular internal dimension at end-systole (LVDs: Control [$n$ = 7] vs. PG-LPS [$n$ = 7]: 2.9 ± 0.07 vs. 3.3 ± 0.06, $P$ < 0.01, **S8B Fig of S2 Data**). Cardiac function remained depressed at 4 weeks after chronic PG-LPS injection (EF: Control [$n$ = 7] vs. PG-LPS [$n$ = 7]: 66 ± 0.5 vs. 57 ± 0.1%, $P$ < 0.01; FS: Control [$n$ = 7] vs. PG-LPS [$n$ = 7]: 32 ± 0.3 vs. 26 ± 0.7%, $P$ < 0.01, **S13A and S14B Figs of S2 Data**). Importantly, PG-LPS-mediated cardiac dysfunction was attenuated by pharmacological inhibition of TLR4 with TAK-242 at 4 weeks (EF: PG-LPS [$n$ = 7] vs. PG-LPS + TAK-242 [$n$ = 7]: 57 ± 1.0 vs. 61 ± 0.6%, $P$ < 0.05; FS: PG-LPS [$n$ = 7] vs. PG-LPS + TAK-242 [$n$ = 7]: 26 ± 0.7 vs. 28 ± 0.4%, $P$ < 0.05, **S13A and S14B Figs of S2 Data**).

These data indicate that chronic PG-LPS infusion at a physiologically realistic dose level induces cardiac dysfunction through the activation of TLR4.

## Effects of PG-LPS on the cardiac fibrosis

We next examined cardiac remodeling, which we evaluated in terms of fibrosis and myocyte apoptosis, because it might lead to progressive heart failure [25,28].

We first evaluated fibrosis by means of Masson-trichrome staining (**Fig 2A and 2B**). Chronic PG-LPS infusion significantly increased the area of fibrosis, compared to the control (Control [$n$ = 7] vs. PG-LPS [$n$ = 7]: 0.9 ± 0.2 vs. 2.9 ± 0.4%, $P$ < 0.01). This PG-LPS-mediated increase of cardiac fibrosis was significantly attenuated by TAK-242 (PG-LPS [$n$ = 7] vs. PG-LPS + TAK-242 [$n$ = 7]: 2.9 ± 0.4 vs. 1.1 ± 0.1%, $P$ < 0.01).

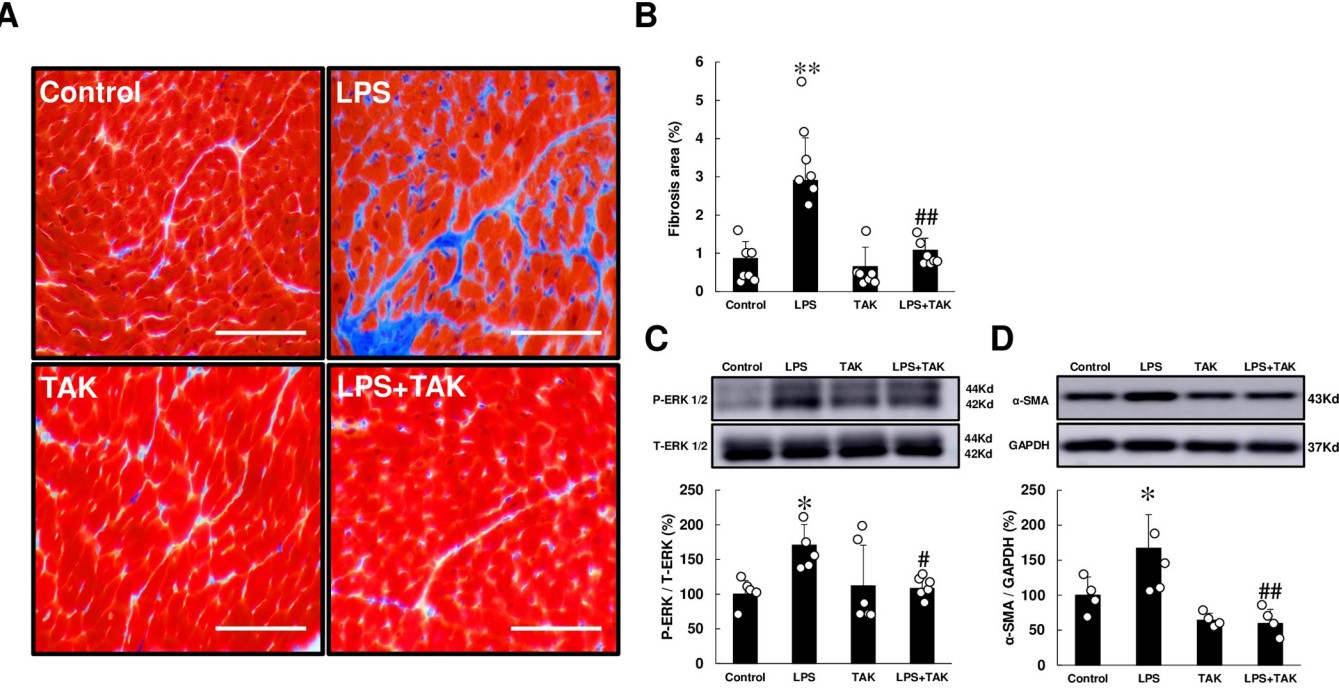

**Fig 2. Effects of chronic PG-LPS infusion on fibrosis in cardiac muscle.** (**A**) Representative images of Masson-trichrome-stained sections of cardiac muscle in the Control (***upper left***), PG-LPS (LPS) (***upper right***), TAK-242 (TAK) (***lower left***) and PG-LPS + TAK-242 (LPS + TAK) (***lower right***) groups. Scale bars: 50 μm. (**B**) The area of fibrosis was significantly increased in the PG-LPS group ($P$ < 0.01 vs. Control), and this increase was significantly attenuated by TAK-242. **\*\***$P$ < 0.01 vs. Control or **##**$P$ < 0.01 vs. PG-LPS group by one-way ANOVA followed by the Tukey-Kramer *post hoc* test. (**C**) Expression of phospho-ERK1/2 (Thr-202/Tyr-204) (**C**) and α-SMA (**D**) was significantly increased in cardiac muscle of PG-LPS-treated mice, and these increases were significantly attenuated by TAK-242. **\***$P$ < 0.05 vs. Control, **#**$P$ < 0.05 vs. PG-LPS group or **##**$P$ < 0.01 vs. PG-LPS group by one-way ANOVA followed by the Tukey-Kramer *post hoc* test. Full-size images of immunoblots are presented in **S3 and S4 Figs of S1 Data**. Data shows means ± SD and scattered dots show individual data.

We also evaluated cardiac fibrosis by measuring the levels of p44/42MAPK (ERK) phosphorylation (**Fig 2C**) and α-SMA expression (**Fig 2D**) at 4 weeks after the start of PG-LPS, because they are closely associated with cardiac fibrosis [18,34]. Expression levels of phospho-ERK (Thr-202/Tyr-204) and α-SMA were significantly increased in cardiac muscle of PG-LPS-treated mice (ERK: Control [$n = 5$] vs. PG-LPS [$n = 5$]: $100 \pm 8.8$ vs. $170 \pm 13\%$, $P < 0.05$; α-SMA: Control [$n = 4$] vs. PG-LPS [$n = 4$]: $100 \pm 13$ vs. $167 \pm 24\%$, $P < 0.05$), and these increases were significantly attenuated by TAK-242 (ERK: PG-LPS [$n = 5$] vs. PG-LPS + TAK-242 [$n = 6$]: $170 \pm 13$ vs. $108 \pm 5.9\%$, $P < 0.05$; α-SMA: Control [$n = 4$] vs. PG-LPS [$n = 4$]: $167 \pm 24$ vs. $59 \pm 10\%$, $P < 0.01$).

These data suggest that cardiac fibrosis induced by chronic PG-LPS infusion might be mediated, at least in part, through the activation of TLR4.

## Effects of PG-LPS on myocyte apoptosis

We next evaluated cardiac myocyte apoptosis by means of terminal deoxyribonucleotidyl transferase (TdT)-mediated biotin-16-deoxyuridine triphosphate (dUTP) nick-end labeling (TUNEL) staining at 4 weeks after the start of PG-LPS (**Fig 3A and 3B**). Chronic PG-LPS infusion increased myocyte apoptosis, compared to the control (Control [$n = 7$] vs. PG-LPS [$n = 7$]: $0.9 \pm 0.1$ vs. $6.6 \pm 1.3\%$, $P < 0.01$). This PG-LPS-mediated increase of cardiac myocyte

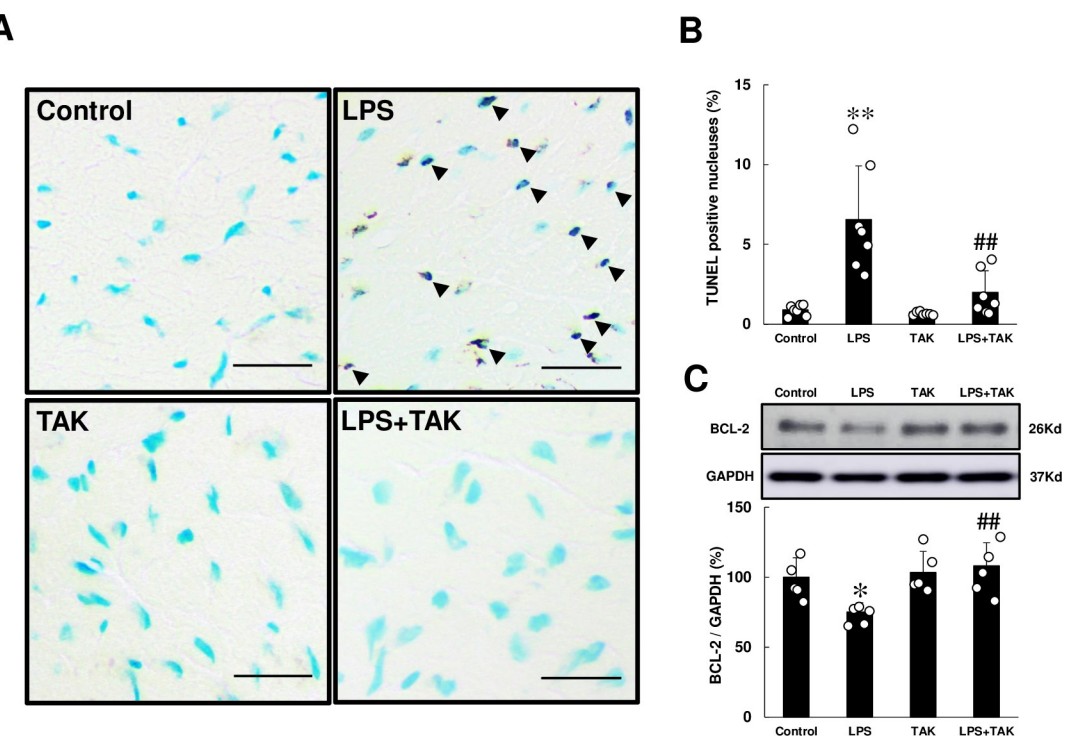

**Fig 3. Effects of chronic PG-LPS infusion on cardiac myocyte apoptosis.** (**A**) Representative images of TUNEL-stained sections of cardiac muscle in the Control (*upper left*), PG-LPS (LPS) (*upper right*), TAK-242 (TAK) (*lower left*) and PG-LPS + TAK-242 (LPS + TAK) (*lower right*) groups. Scale bars: 2 μm. Black arrows: TUNEL-positive myocytes. (**B**) The number of TUNEL-positive myocyte was significantly increased in the PG-LPS group ($P < 0.01$ vs. Control), and this increase was significantly attenuated by TAK-242. $^{**}P < 0.01$ vs. Control or $^{\#\#}P < 0.01$ vs. PG-LPS group by one-way ANOVA followed by the Tukey-Kramer *post hoc* test. (**C**) Expression of anti-apoptotic BCL-2 protein was significantly decreased in cardiac muscle of PG-LPS-treated mice, and this change was significantly attenuated by TAK-242. $^{*}P < 0.05$ vs. Control or $^{\#\#}P < 0.01$ vs. PG-LPS group by one-way ANOVA followed by the Tukey-Kramer *post hoc* test. Full-size images of immunoblots are presented in **S5 Fig of S1 Data**. Data shows means ± SD and scattered dots show individual data.

apoptosis was attenuated by TAK-242 (PG-LPS [$n = 7$] vs. PG-LPS + TAK-242 [$n = 7$]: 6.6 ± 1.3 vs. 2.0 ± 0.5%, $P < 0.01$).

We also evaluated cardiac myocyte apoptosis by measuring the change of anti-apoptotic BCL-2 protein in the heart (Fig 3C). B cell lymphoma 2 (BCL-2) expression was significantly decreased in cardiac muscle of PG-LPS-treated mice (Control [$n = 5$] vs. PG-LPS [$n = 5$]: 100 ± 6.2 vs. 75 ± 2.9%, $P < 0.05$) and the increase was significantly attenuated by TAK-242 (PG-LPS [$n = 5$] vs. PG-LPS + TAK-242 [$n = 5$]: 75 ± 2.9 vs. 108 ± 4.0%, $P < 0.01$).

These data suggest that the induction of cardiac myocyte apoptosis induced by chronic PG-LPS infusion might be mediated, at least in part, through the activation of TLR4.

## Oxidative stress was increased by chronic PG-LPS infusion

TLR4 signaling activates various downstream signal transduction pathways, including oxidative stress, by increasing the generation of reactive oxygen species (ROS) [35].

We thus examined in-situ expression of 8-hydroxy-2'-deoxyguanosine (8-OHdG), a DNA modification product generated by ROS, as a marker for oxidative DNA damage in the four groups (Fig 4A and 4B). We examined the validity of the 8-OHdG immunostaining used in this study by incubation in TBS-T with (positive control) or without (negative control) 0.3% $H_2O_2$ at room temperature for 1 h prior to anti-8-OHdG antibody treatment. The results demonstrate that the 8-OHdG staining procedure clearly distinguished 8-OHdG-positive from non-positive nuclei (S2 Fig of S1 Data).

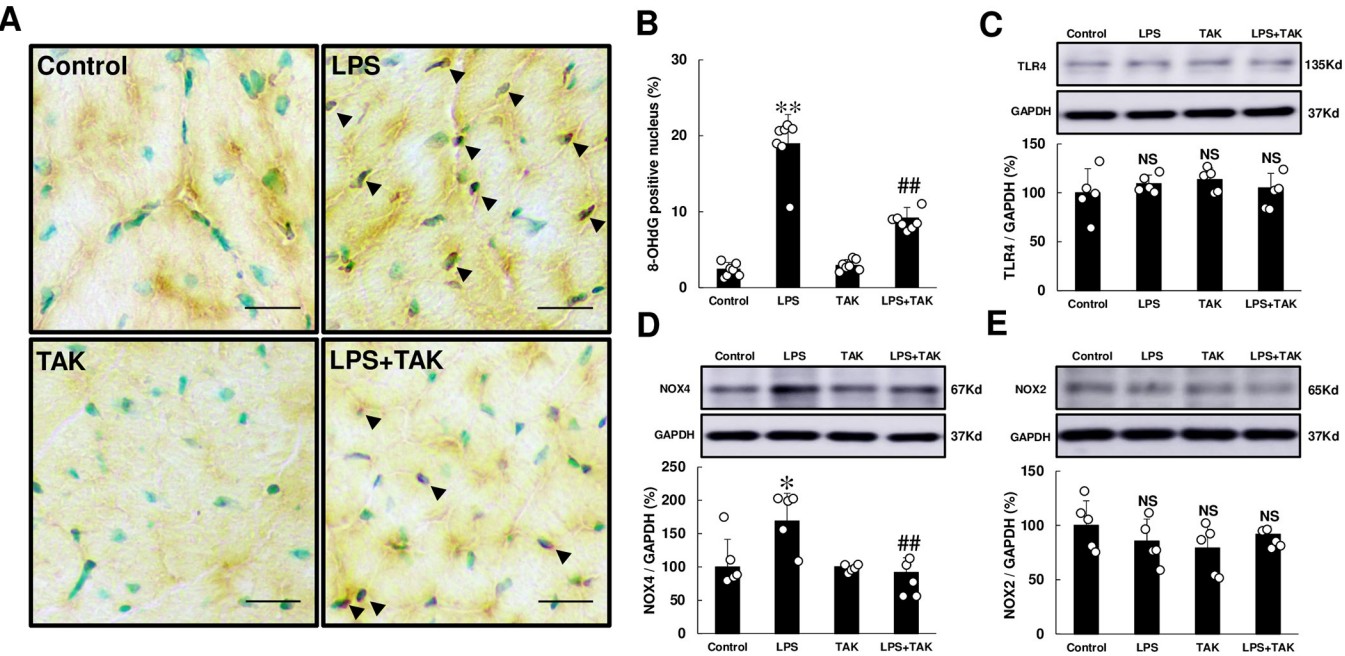

**Fig 4. Effects of chronic PG-LPS infusion on oxidative stress. (A)** Representative images of 8-OHdG-immunostained sections of cardiac muscle in the Control (*upper left*), PG-LPS (LPS) (*upper right*), TAK-242 (TAK) (*lower left*) and PG-LPS + TAK-242 (LPS + TAK) (*lower right*) groups. Scale bars: 2 μm. Black arrows: TUNEL-positive myocytes. **(B)** The number of 8-OHdG-positive myocytes was significantly increased in the PG-LPS group, and this increase was significantly attenuated by TAK-242. **P < 0.01 vs. Control or ##P < 0.01 vs. PG-LPS group by one-way ANOVA followed by the Tukey-Kramer *post hoc* test. **(C)** TLR4 expression was similar among the four groups. NS, not significantly different from the Control ($P > 0.05$) by one-way ANOVA followed by the Tukey-Kramer *post hoc* test. Full-size images of immunoblots are presented in **S6 Fig of S1 Data**. **(D)** NOX4 expression was significantly increased in the PG-LPS group, and this increase was significantly attenuated by TAK-242. *P < 0.05 vs. Control or ##P < 0.01 vs. PG-LPS group by one-way ANOVA followed by the Tukey-Kramer *post hoc* test. Full-size images of immunoblots are presented in **S7 Fig of S1 Data**. **(E)** NOX2 expression was similar among the four groups. NS, not significantly different from the Control ($P > 0.05$) by one-way ANOVA followed by the Tukey-Kramer *post hoc* test. Full-size images of immunoblots are presented in **S8 Fig of S1 Data**. Data shows means ± SD and scattered dots show individual data.

The number of 8-OHdG-positive cardiac myocytes was significantly increased at 4 weeks after PG-LPS treatment (Control [$n = 7$] vs. PG-LPS [$n = 7$]: 2.5 ± 0.3 vs. 19.0 ± 1.4%, $P < 0.01$), but this increase was significantly attenuated by TAK-242 (PG-LPS [$n = 7$] vs. PG-LPS + TAK-242 [$n = 7$]: 19.0 ± 1.4 vs. 9.2 ± 0.5%, $P < 0.01$) (**Fig 4B**).

These data suggest that oxidative stress might also play an important role, at least in part, in the development of PG-LPS-mediated cardiac dysfunction via TLR4.

## NOX4 expression was increased by chronic PG-LPS infusion

Nicotinamide adenine dinucleotide phosphate oxidase-4 (NOX4) is the only known enzymes, whose physiological function is to produce ROS in cardiac myocytes [36]. NOX2 and NOX4 are abundantly expressed in cardiac myocytes, and their activity is controlled by their expression level [35]. We thus examined NOX2 and NOX4 expression in the heart in the four groups.

TLR4 expression was similar among the four groups (**Fig 4C**). However, NOX4 expression was significantly increased (Control [$n = 5$] vs. PG-LPS [$n = 5$]: 100 ± 18 vs. 169 ± 19%, $P < 0.05$). This increase was significantly attenuated by TAK-242 (PG-LPS [$n = 5$] vs. PG-LPS + TAK-242 [$n = 5$]: 169 ± 19 vs. 92 ± 10%, $P < 0.01$) (**Fig 4D**). In contrast, NOX2 expression was similar among the four groups (**Fig 4E**).

These data suggest that chronic PG-LPS infusion increased the generation of ROS via activation of the TLR4-NOX4 pathway.

## Necroptosis was increased by chronic PG-LPS infusion

ROS derived from NOX4 was recently demonstrated to cause apoptosis and necroptosis via activation of receptor-interacting protein 3 (RIP3), a key determinant of programmed necrosis (necroptosis), in endothelial cells [37]. We thus examined the amount of phospho-RIP3 (Thr-231/Ser-232) and found that it was significantly increased in the heart of PG-LPS-treated mice (Control [$n = 5$] vs. PG-LPS [$n = 4$]: 100 ± 6.8 vs. 177 ± 15%, $P < 0.05$). Again, this increase was significantly attenuated by TAK-242 (PG-LPS [$n = 4$] vs. PG-LPS + TAK-242 [$n = 5$]: 177 ± 15 vs. 86 ± 21%, $P < 0.01$, $n = 5$) (**Fig 5A**).

These data suggest that chronic PG-LPS infusion might increase RIP3 phosphorylation, leading to apoptosis and necroptosis, via ROS production derived from TLR4-NOX4 signaling in cardiac muscle.

## Phosphorylation and oxidation of CaMKII was increased by chronic PG-LPS infusion

Calmodulin kinase II (CaMKII) was recently demonstrated to be one of the targets of RIP3, which activates CaMKII via phosphorylation and oxidation [38]. We thus examined the amounts of phospho-CaMKII (Thr-286) (**Fig 5B**) and oxidized methionine-281/282 CaMKII (ox-CaMKII) (**Fig 5C**) in the heart of PG-LPS-treated mice and found that they were significantly increased at 4 weeks after the PG-LPS treatment (CaMKII (Thr-286): Control [$n = 6$] vs. PG-LPS [$n = 5$]: 100 ± 12.1 vs. 213 ± 25%, $P < 0.05$; ox-CaMKII: Control [$n = 4$] vs. PG-LPS [$n = 4$]: 100 ± 20 vs. 412 ± 72%, $P < 0.01$). These changes were significantly attenuated by TAK-242 (CaMKII [Thr-286]: PG-LPS [$n = 5$] vs. PG-LPS + TAK-242 [$n = 4$]: 213 ± 25 vs. 103 ± 31%, $P < 0.05$; ox-CaMKII: PG-LPS [$n = 4$] vs, PG-LPS + TAK-242 [$n = 4$]: 412 ± 72 vs. 186 ± 19%, $P < 0.05$).

These data suggest that chronic PG-LPS infusion might activate TLR4/RIP3/CaMKII signaling in cardiac myocytes.

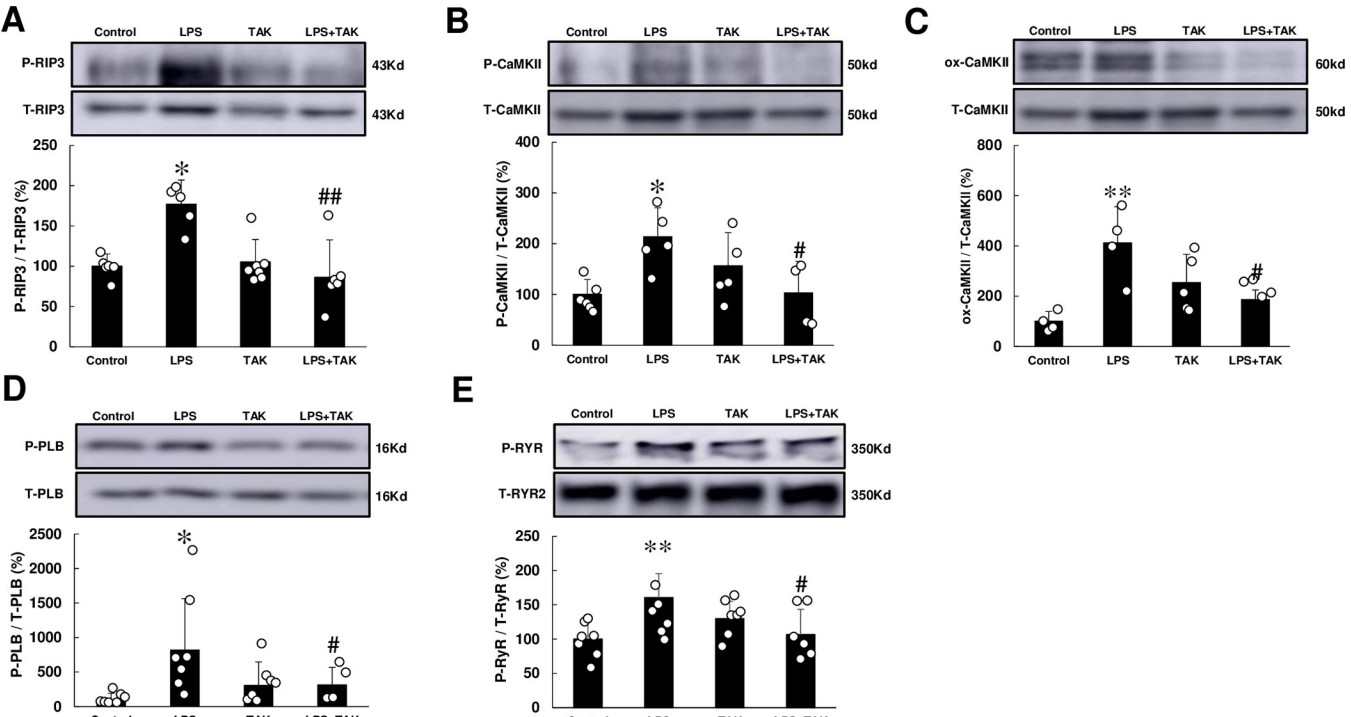

**Fig 5. Effects of chronic PG-LPS infusion on RIP3, CaMKII, PLN, RyR2 in the heart. (A)** RIP3 phosphorylation (Thr-231/Ser-232) was significantly increased in the PG-LPS group, and this increase was significantly attenuated by TAK-242. $^*P < 0.05$ vs. Control or $^{##}P < 0.01$ vs. PG-LPS group by one-way ANOVA followed by the Tukey-Kramer *post hoc* test. Full-size images of immunoblots are presented in **S9 Fig of S1 Data. (B)** CaMKII phosphorylation (Thr-286) was significantly increased in the PG-LPS group, and this increase was significantly attenuated by TAK-242. $^*P < 0.05$ vs. Control or $^{##}P < 0.01$ vs. PG-LPS group by one-way ANOVA followed by the Tukey-Kramer *post hoc* test. Full-size images of immunoblots are presented in **S10 Fig of S1 Data. (C)** CaMKII oxidation was significantly increased in the PG-LPS group, and this increase was significantly attenuated by TAK-242. $^{**}P < 0.01$ vs. Control or $^#P < 0.05$ vs. PG-LPS group by one-way ANOVA followed by the Tukey-Kramer *post hoc* test. Full-size images of immunoblots are presented in **S11 Fig of S1 Data. (D)** PLN phosphorylation (Thr-17) was significantly increased in the PG-LPS group, and this increase was significantly attenuated by TAK-242. $^{**}P < 0.05$ vs. Control or $^#P < 0.05$ vs. PG-LPS group by one-way ANOVA followed by the Tukey-Kramer *post hoc* test. Full-size images of immunoblots are presented in **S12 Fig of S1 Data. (E)** RyR2 phosphorylation (Ser-2814) was significantly increased in the PG-LPS group, and this increase was significantly attenuated by TAK-242. $^{**}P < 0.01$ vs. Control or $^#P < 0.05$ vs. PG-LPS group by one-way ANOVA followed by the Tukey-Kramer *post hoc* test. Full-size images of immunoblots are presented in **S13 Fig of S1 Data.** Data show means ± SD and representative immunoblotting are shown.

## Phosphorylation of PLN and RyR2 was increased by chronic PG-LPS infusion

ROS derived from NOX4 was recently demonstrated to induce $Ca^{2+}$-mishandling via altered phosphorylation of phospholamban (PLN) on Thr-17 and ryanodine receptor 2 (RyR2) on Ser-2814, mediated by CaMKII as well as ox-CaMKII, leading to cardiac remodeling and dysfunction [39]. We thus examined the effects of chronic PG-LPS on phosphorylation of PLN, focusing on Thr-17, which is phosphorylated by CaMKII (**Fig 5D**).

Phospho-PLN (Thr-17) was significantly increased in cardiac muscle of PG-LPS- treated mice (Control [*n* = 7] vs. PG-LPS [*n* = 7]: 100 ± 34 vs. 824 ± 279%, *P* < 0.05 vs. Control) (**Fig 5D**) and this increase was significantly attenuated by TAK-242 (PG-LPS [*n* = 7] vs. PG-LPS + TAK-242 [*n* = 4]: 824 ± 279 vs. 320 ± 123%, *P* < 0.05).

Like PLN, phospho-RyR2 (Ser-2814), which is phosphorylated by CaMKII, was also significantly increased in cardiac muscle of PG-LPS- treated mice (Control [*n* = 7] vs. PG-LPS [*n* = 6]: 100 ± 9.6 vs. 161 ± 14.2%, *P* < 0.05 vs. Control) (**Fig 5E**) and this increase was significantly attenuated by TAK-242 (PG-LPS [*n* = 6] vs. PG-LPS + TAK-242 [*n* = 6]: 161 ± 14.2 vs. 107 ± 15%, *P* < 0.05).

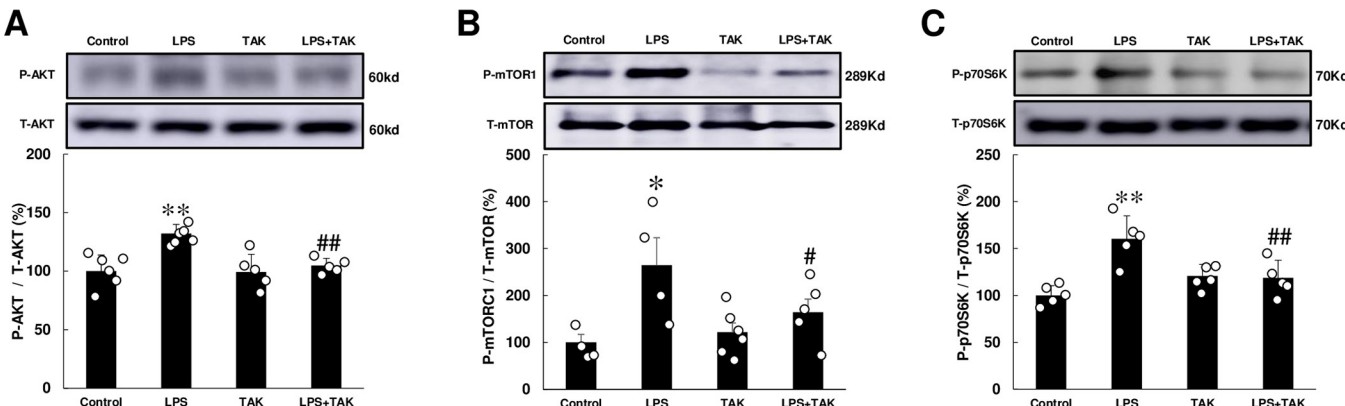

**Fig 6. Effects of chronic PG-LPS infusion on Akt, mTOR, p70S6K, LC3, p62 in the heart. (A)** Akt phosphorylation (Ser-473) was significantly increased in the PG-LPS group, and this increase was significantly attenuated by TAK-242. **P < 0.01 vs. Control or ##P < 0.01 vs. PG-LPS group by one-way ANOVA followed by the Tukey-Kramer *post hoc* test. Full-size images of immunoblots are presented in **S14 Fig of S1 Data. (B)** mTORC1 phosphorylation (Ser-2448) was significantly increased in the PG-LPS group, and this increase was significantly attenuated by TAK-242. *P < 0.05 vs. Control or ##P < 0.05 vs. PG-LPS group by one-way ANOVA followed by the Tukey-Kramer *post hoc* test. Full-size images of immunoblots are presented in **S15 Fig of S1 Data. (C)** p70S6K phosphorylation (Ser-389) was significantly increased in the PG-LPS group, and this increase was significantly attenuated by TAK-242. **P < 0.01 vs. Control or ##P < 0.01 vs. PG-LPS group by one-way ANOVA followed by the Tukey-Kramer *post hoc* test. Full-size images of immunoblots are presented in **S16 Fig of S1 Data.** Data shows means ± SD and scattered dots show individual data.

These data suggest that chronic PG-LPS infusion might increase phosphorylation of PLN and RyR2 via TLR4-NOX4 signaling, leading to $Ca^{2+}$-mishandling in cardiac myocytes.

## Akt-mTOR-p70S6K signaling was activated by chronic PG-LPS infusion

NOX4 activation was recently demonstrated to cause cardiac remodeling through activation of Akt- mechanistic target of rapamycin (mTOR)-p70 ribosomal S6 kinase (p70S6K) signaling [40].

Phospho-Akt (Ser-473) was significantly increased in the heart of PG-LPS-treated mice at 4 weeks (Control [$n = 6$] vs. PG-LPS [$n = 6$]: 100 ± 5.6 vs. 132 ± 3.1%, $P < 0.01$), and this increase was significantly attenuated by TAK-242 (PG-LPS [$n = 6$] vs. PG-LPS + TAK-242 [$n = 5$]: 132 ± 3.1 vs. 105 ± 3.1, $P < 0.01$) (Fig 6A).

Phospho-mTORC1 (Ser-2448) and phospho-p70S6K (Thr-389) were similarly significantly increased in the heart of PG-LPS-treated mice at 4 weeks (mTORC1: Control [$n = 4$] vs. PG-LPS [$n = 4$]: 100 ± 17.3 vs. 264 ± 58.4%, $P < 0.05$; p70S6K: Control [$n = 5$] vs. PG-LPS [$n = 5$]: 100 ± 4.7 vs. 160 ± 11%, $P < 0.01$). These increases were significantly attenuated by TAK-242 (mTORC1: PG-LPS [$n = 4$] vs. PG-LPS + TAK-242 [$n = 5$]: 264 ± 58.4 vs. 164 ± 28.3%, $P < 0.05$; p70S6K: PG-LPS [$n = 5$] vs. PG-LPS + TAK-242 [$n = 5$]: 160 ± 11 vs. 119 ± 8.4%, $P < 0.01$) (Fig 6B and 6C).

These data suggest that chronic PG-LPS infusion might increase the activation of Akt-mTOR-p70S6K signaling, thereby contributing to cardiac remodeling and cardiac dysfunction.

## Autophagic activity was increased by chronic PG-LPS infusion

We next investigated the effects of PG-LPS on autophagy in the cardiac muscle (Fig 7A), because TLR4-NOX4 signaling is known to promote cell death through autophagy in rats with heart failure induced by aortic banding [41].

The amount of microtubule-associated protein 1 light chain 3 (LC3)-II was measured in terms of LC3-II/LC3-I, which correlates with the number of autophagosomes and provides a

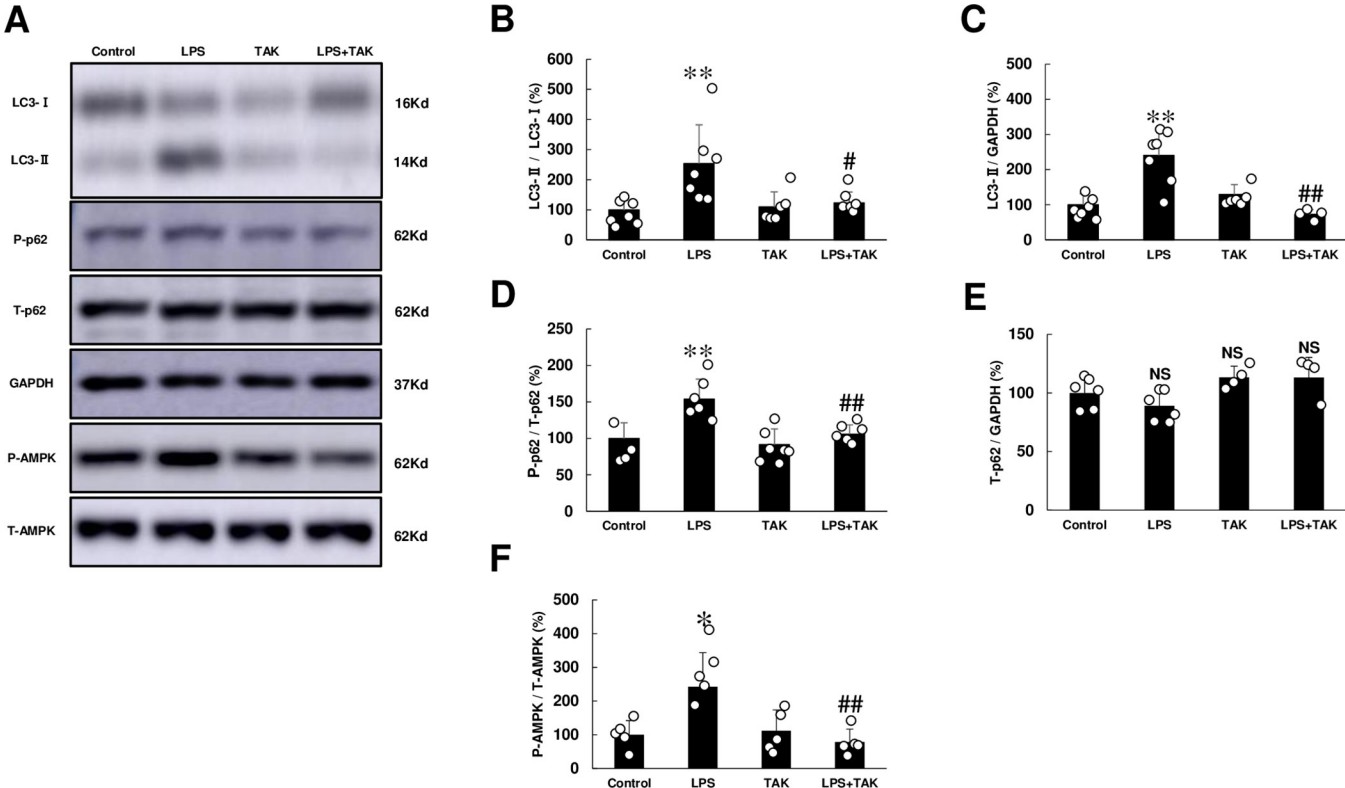

**Fig 7. (A)** Representative immunoblotting of LC3-I, LC3-II, phospho-p62 (Ser-351), p62, AMPK and GAPDH are shown. Full-size images of immunoblots are presented in **S17-S21 Figs of S1 Data**. **(B)** LC3-II/LC3-I was significantly increased in the PG-LPS group, and this increase was significantly attenuated by TAK-242. **P < 0.01 vs. Control or #P < 0.05 vs. PG-LPS group by one-way ANOVA followed by the Tukey-Kramer *post hoc* test. **(C)** LC3-II expression was significantly increased in the PG-LPS group, and this increase was significantly attenuated by TAK-242. **P < 0.01 vs. Control or #P < 0.05 vs. PG-LPS group by one-way ANOVA followed by the Tukey-Kramer *post hoc* test. **(D)** p62 phosphorylation (Ser-351) was significantly increased in the PG-LPS group, and this increase was significantly attenuated by TAK-242. **P < 0.01 vs. Control or #P < 0.05 vs. PG-LPS group by one-way ANOVA followed by the Tukey-Kramer *post hoc* test. **(E)** p62 expression was similar among the four groups. NS, not significantly different from the Control (P > 0.05) by one-way ANOVA followed by the Tukey-Kramer *post hoc* test. **(F)** AMPK (Thr-172) was significantly increased in the PG-LPS group, and this increase was significantly attenuated by TAK-242. *P < 0.05 vs. Control or ##P < 0.01 vs. PG-LPS group by one-way ANOVA followed by the Tukey-Kramer *post hoc* test. Data are means ± SD and representative immunoblots are shown. Data shows means ± SD and scattered dots show individual data.

good index of autophagy induction [42]. LC3-II/LC3-I was significantly increased in the heart of PG-LPS-treated mice for 4 weeks (Control [*n* = 7] vs. PG-LPS [*n* = 7]: 100 ± 11.7 vs. 254 ± 49%, *P* < 0.01). This increase was significantly attenuated by TAK-242 (PG-LPS [*n* = 7] vs. PG-LPS + TAK-242 [*n* = 6]: 254 ± 49 vs. 123 ± 15%, *P* < 0.05) (**Fig 7B**). We also examined LC3-II expression and found that it was significantly increased in the heart of PG-LPS-treated mice at 4 weeks (Control [*n* = 7] vs. PG-LPS [*n* = 7]: 100 ± 12 vs. 240 ± 29%, *P* < 0.01). The increase was significantly attenuated by TAK-242 (PG-LPS [*n* = 7] vs. PG-LPS + TAK-242 [*n* = 4]: 240 ± 29 vs. 73 ± 7%, *P* < 0.01) (**Fig 7C**).

Increased LC-3-II could reflect either increased autophagosome formation due to autophagy induction, or a blockage in the downstream steps from autophagy, such as insufficient fusion or decreased autophagosome degradation [43]. Increased p62 phosphorylation on serine 351 is known to be necessary for autophagic degradation of polyubiquitinated proteins and for recruiting autophagy machinery proteins [44]. Furthermore, mTOR phosphorylation (Ser-2448) was recently demonstrated to increase the phosphorylation of serine 351 of p62 [45]. We thus examined the level of p62 expression as well as phosphorylation on serine 351, and found that p62 phosphorylation was significantly increased in the PG-LPS group (Control [*n* = 4] vs.

PG-LPS [$n = 6$]: $100 \pm 11$ vs. $154 \pm 11\%$, $P < 0.01$). This increase was suppressed by TAK-242 (PG-LPS [$n = 6$] vs. PG-LPS + TAK-242 [$n = 6$]: $154 \pm 11$ vs. $106 \pm 5$, $P < 0.01$ vs. PG-LPS) (**Fig 7D**). However, p62 expression was similar among all four groups (**Fig 7E**).

These data suggest that chronic PG-LPS treatment might increase autophagic activity via TLR4 activation, leading to an increase of autophagosome number as well as autophagy flux.

## AMPK signaling was activated by chronic PG-LPS infusion

Cardiac excitation-contraction coupling is impaired by enhancing sarcoplasmic reticulum $Ca^{2+}$ leakage through TLR4-ROS signaling via CaMKII phosphorylation and oxidization [46]. AMP-activated protein kinase (AMPK), which might lie downstream of CaMKII and can be phosphorylated on Thr-172 in response to an increased intracellular $Ca^{2+}$ level, was recently demonstrated to regulate a variety of metabolic processes including autophagy [47,48]. We thus examined the effects of chronic PG-LPS activation on phosphorylation of AMPK.

Phospho-AMPK (Thr-172) was significantly increased in cardiac muscle of PG-LPS-treated mice at 4 weeks (Control [$n = 5$] vs. PG-LPS [$n = 5$]: $100 \pm 19$ vs. $243 \pm 45\%$, $P < 0.05$) and this increase was significantly attenuated by TAK-242 (PG-LPS [$n = 5$] vs. PG-LPS + TAK-242 [$n = 5$]: $243 \pm 45$ vs. $79 \pm 17\%$, $P < 0.01$) (**Fig 7F**).

These data suggest that chronic PG-LPS treatment might increase autophagic activity though activation of CaMKII-AMPK signaling via TLR4.

## Discussion

Our findings here indicate that cardiac function was significantly impaired in mice treated with PG-LPS at a dose consistent with circulating levels in PD patients, and myocyte apoptosis, fibrosis and oxidative stress were significantly increased. Importantly, these changes were blunted by pharmacological inhibition of TLR4 with TAK-242. We then investigated the mechanism of these changes.

The pro-inflammatory effect of PG-LPS on macrophages is controversial. Martin et al. reported that 1 ng/ml *E. Coli* LPS and 10.000 ng/ml PG-LPS induced similar levels of IL-6 and TNF-α in a human macrophage cell line, THP-1 cells [49], and another group also found that PG-LPS is a less potent inducer of pro-inflammatory cytokines than *E. Coli* LPS [50]. On the other hand, Jones et al reported that PG-LPS induced a strong pro-inflammatory cytokine response, particularly for IL-1β, IL-6, and MCP-1, in a mouse alveolar macrophage cell line AMJ2-C8, compared to the *E.coli* LPS [7]. Therefore, we examined the effects of chronic PG-LPS infusion on serum levels of IL-6, IL-1β, IL-10, IL-17, IFN-γ, MCP-1 and TNF-α, in addition to endothelial biomarkers PDGF-BB and VEGF, in order to exclude the possibility that the cardiac dysfunction observed here might be simply a consequence of inflammation caused by chronic PG-LPS infusion [51,52]. However, we found no significant differences among the four groups in this study.

VEGF and PDGF can cause endothelial dysfunction, which might induce matrix remodeling and cardiac dysfunction [53]. However, in the present model, the serum levels of VEGF and PDGF-BB were similar among the four groups, suggesting that PG-LPS does not induce endothelial dysfunction at the dose used in this study.

We thus focused on the TLR4 and its downstream signaling to clarify the mechanism of PG-LPS-mediated cardiac dysfunction, and hypothesized that ROS production and oxidative stress, via activation of the TLR4-NOX4 pathway, might play a key role.

NOX2 and NOX4 are the predominant isoforms expressed in cardiac myocytes [54]. These enzymes exhibit high sequence homology, but have distinct characteristics. NOX2 is localized primarily in the plasma membrane, and its activation contributes to cardiac hypertrophy,

fibrosis and heart failure via increased oxidative stress [55]. In addition, ROS derived from TLR4-NOX2 signaling is known to be involved in killing and degrading microorganisms, contributing to host defense function in patients with infectious diseases [56,57]. Therefore, we anticipated that NOX2 expression might be increased in the heart of PG-LPS-treated mice, but in fact we found NOX2 expression was similar among the four groups, as shown in **Fig 4E**. We do not know why NOX2 expression was not increased in the heart after PG-LPS infusion. PG was recently reported to circumvent the NOX2-mediated antimicrobial defense system, potentially enabling persistence in the tissue that supports the teeth [58], and so it is possible that PG-mediated circumvention of the host defense system via NOX2 might occur in the heart of PG-LPS-treated mice.

NOX4 is found in intracellular membranes, such as mitochondria, endoplasmic reticulum, nuclear and sarcoplasmic reticulum membranes [59]. NOX4 is constitutively active and is regulated primarily at the level of its expression [59]. In this study, we found that NOX4 expression was significantly increased in the heart of PG-LPS-treated mice. Notably, this increase of NOX4 expression was effectively alleviated by pharmacological inhibition of TLR4 with TAK-242. These results suggest that PG-LPS activates TLR4-NOX4 signaling in the heart.

Little is known regarding the isoform-specific role of NOX in the development of PG-LPS-mediated cardiac remodeling. Emerging data are conflicting, with considerable debate as to whether NOX4 is protective [60] or deleterious [40] for the development of cardiac remodeling. In this study, the impairment of cardiac function in PG-LPS-treated mice was associated with activation of the TLR4-NOX4 pathway, because cardiac fibrosis, cardiac myocyte apoptosis, oxidative stress and autophagy were all significantly increased in the heart of PG-LPS-treated mice, and these changes were attenuated by the TRL4 receptor antagonist TAK-242.

In addition, we could not exclude the involvement of ROS derived from NOX1, a minor NOX subtype in the heart, in the development of PG-LPS-mediated cardiac dysfunction, because NOX1 might play a role in cardiac myocyte apoptosis after exposure to *Escherichia coli* LPS [61,62]. Also, *Helicobacter pylori* LPS induces ROS through the activation of TLR4-NOX1 signaling, leading to apoptosis of guinea pig gastric mucosal cells and gastric ulcer formation [63]. These data might suggest that NOX1 is involved in the process of PG-LPS-mediated cardiac dysfunction. However, we think this is unlikely, because NOX1-mediated cardiac myocyte apoptosis is induced through decreased Akt phosphorylation in the heart of *Escherichia coli* LPS treated mice [62], which is in contrast to the increased Akt phosphorylation in the heart of PG-LPS-treated mice (**Fig 6A**).

We found that CaMKII-mediated PLN phosphorylation (Thr-17) and RyR2 phosphorylation (Ser-2448) were significantly increased in the heart of PG-LPS-treated mice, and TAK-242 ameliorated these changes. It has been reported that NOX4 in the sarcoplasmic reticulum regulates RyR1 phosphorylation and intracellular $Ca^{2+}$ leakage in skeletal muscle [64]. Our data, together with the previous findings, suggest that local ROS production derived from NOX4 in the sarcoplasmic reticulum might be associated with the increased PLN and RyR2 phosphorylation via activation of CaMKII.

It has been reported that ROS derived from NOX4 in the mitochondrial membrane might cause mitochondrial dysfunction and cardiac myocyte apoptosis during aging and heart failure [65]. In this study, we found that cardiac myocyte apoptosis was significantly increased in the heart of PG-LPS-treated mice and this increase was attenuated by TAK-242. Our current findings, together with the previous studies, suggested that local ROS production via NOX4 in the mitochondrial membrane might be involved, at least in part, in the induction of cardiac apoptosis by chronic PG-LPS infusion.

The TLR4-NOX4 pathway was demonstrated to cause cardiac fibrosis and hypertrophy by enhancing Akt-mTOR signaling [40]. mTOR has been found at various locations, including

lysosomes, nuclei, mitochondria and plasma membrane [66]. Although it remains unclear which localization of NOX4 within cardiac myocytes might be involved in activating the Akt-mTOR-p70S6K pathway, we confirmed that Akt-mTOR-p70S6K signaling was activated in the heart of PG-LPS-treated mice, and this might contribute to cardiac remodeling.

Activation of the TLR4-NOX4 pathway is closely related to autophagy, and ROS generated via NOX4 within the endoplasmic reticulum plays an important role in this process [59]. Activation of TLR4-NOX4 pathway was also reported to promote cell death through autophagy during the progression of heart failure [41]. We thus examined the autophagic activity in terms of the LC3-II/LC3-I, which correlate with the number of autophagosomes [42], and p62 phosphorylation on serine 351, an indicator of autophagy flux [44]. Both the LC3-II/LC3-I ratio and the level of p62 phosphorylation on serine 351 were increased in the heart of PG-LPS-treated mice at 4 weeks, and these increases were blocked by TAK-242. These findings, together with the previous studies, suggest that increased autophagic activity due to activation of the TLR4-NOX pathway might contribute to PG-LPS-mediated cardiac remodeling.

## Conclusion

Immune system activation induced by chronic PG-LPS infusion contributes to the development of myocardial remodeling and cardiac dysfunction via activation of TLR4 signaling. Furthermore, activation of the TLR4-NOX4 pathway, which was previously reported to induce heart failure [67,68], might play a role in PG-LPS-mediated cardiac dysfunction. Consequently, pharmacological inhibition of the TLR4-NOX4 pathway might be a promising strategy to decrease ROS production in patients with periodontitis.

## Supporting information

**S1 Data.**
(PDF)

**S2 Data.**
(PDF)

## Author Contributions

**Conceptualization:** Ichiro Matsuo, Naoya Kawamura, Yoshiki Ohnuki, Kenji Suita, Satoshi Okumura.

**Formal analysis:** Ichiro Matsuo, Naoya Kawamura, Kenji Suita, Takehiro Matsubara, Satoshi Okumura.

**Funding acquisition:** Ichiro Matsuo, Naoya Kawamura, Yoshiki Ohnuki, Kenji Suita, Aiko Ito, Megumi Nariyama, Satoshi Okumura.

**Investigation:** Ichiro Matsuo, Naoya Kawamura, Kenji Suita, Takehiro Matsubara, Aiko Ito, Yoshio Hayakawa, Akinaka Morii, Kenichi Kiyomoto, Michinori Tsunoda.

**Methodology:** Ichiro Matsuo, Naoya Kawamura, Yoshiki Ohnuki, Kenji Suita, Misao Ishikawa, Takehiro Matsubara, Yasumasa Mototani.

**Supervision:** Kazuhiro Gomi, Satoshi Okumura.

**Writing – original draft:** Satoshi Okumura.

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
