## [Decision Letter · Decision Letter 0]

8 Mar 2022

PONE-D-21-32097Role of TLR4 signaling on Porphyromonas gingivalis LPS-induced cardiac dysfunction in micePLOS ONE

Dear Dr. Okumura,

Thank you for submitting your manuscript to PLOS ONE. After careful consideration, we feel that it has merit but does not fully meet PLOS ONE’s publication criteria as it currently stands. Therefore, we invite you to submit a revised version of the manuscript that addresses the points raised during the review process.

Your paper was reviewed by two experts in the field. Both reviewers support your conclusion, but some minor modifications are necessary. Please read the comments and address the issues accordingly.

We look forward to receiving your revised manuscript.

Kind regards,

Tomohiko Ai, M.D., Ph.D.

Academic Editor

PLOS ONE

Journal Requirements:

Reviewers' comments:

Reviewer's Responses to Questions

**Comments to the Author**

1. Is the manuscript technically sound, and do the data support the conclusions?

Reviewer #1: Yes

Reviewer #2: Yes

2. Has the statistical analysis been performed appropriately and rigorously? 

Reviewer #1: Yes

Reviewer #2: Yes

3. Have the authors made all data underlying the findings in their manuscript fully available?

Reviewer #1: Yes

Reviewer #2: Yes

4. Is the manuscript presented in an intelligible fashion and written in standard English?

Reviewer #1: Yes

Reviewer #2: Yes

5. Review Comments to the Author

Reviewer #1: In this paper, the authors have focused on NOX2 and NOX4, which are expressed in cardiac myocytes among several NOX subunits. This choice makes sense. In particular, the authors focused on NOX4, which is highly responsive to LPS stimulation and provides an excellent research data and discussion of TLR4-NOX4 signalling in cardiac myocytes . 1) However, in cardiomyocytes, not only NOX4 but also NOX1 / NADPH oxidase is strongly involved in endotoxin-induced cardiac myocytes apoptosis. Therefore, in the Discussion section, I recommend you to make some inferences about the relationship between NOX1 and NOX4. 2) Similarly, please mention briefly here that no LPS-TLR4-NOX2 signalling was observed in cardiac myocytes as shown in Figure 4E. Since NOX2-derived ROS kills and degrades bacteria, there may be a reason why NOX2 is not responsive to LPS-TLR4 stimulation in cardiac myocytes.

P.S. In the description of Trichrome Statin Kit (#HT15-1KT; Sigma-Aldrich, St. Louis, MO); USA is required after MO in parentheses.

Reviewer #2: Thank you for submitting to PLOS ONE.

I’m very interested in your article “Role of TLR4 signaling on Porphyromonas gingivalis LPS-induced cardiac dysfunction in mice”.

I recognize that the main points of this paper are the following two.

Immune system activation induced by Lipopolysaccharide on gram-negative bacterial pathogen known to be the cause of periodontitis, induce cardiac dysfunction and myocardial remodeling via the TLR4 signal. Furthermore, it was suggested that the TLR4-NOX4 pathway previously reported in the heart failure model may play an important role.

I sound that this paper is suitable for publication in PLOS ONE

However, one major revisions and some minor revisions are required.

Major revisions：

The target of your study is myocardial cell damage and cardiac dysfunction. However, cardiovascular events mediated by arteriosclerosis are discussed in introduction. Originally, if you would consider cardiovascular events related to atherosclerosis, coronary artery or vessels must be also investigated. I think introduction should focus on myocardial cell damage. For example, how about changing the reference 1) 12) to other paper such as heart failure model related to myocardial cell damage and discussing it?

Minor revisions：

1) You should write P-value at every item in all tables, even if there is no significant difference.

2) The order and abbereviation of the items in the Table2 A&B have to be changed as follows,

Order

1 EF

2 LVEDV

3 ESV → LVESV

4 FS

5 LVIDd →LVDd

6 LVIDs →LVDs

7 HR

8 SV

9 CO

10 IVSDd

11 LVPWTd →PWTd

Moreover, except LVSTs＆LVPWTs, and change Abbereviation details below.

EF (%): ejection fraction

LVEDV (mL): left ventricular end-diastolic volume

LVESV (mL): left ventricular end-systolic volume

FS (%): fractional shortening

LVDd (mm): left ventricular dimension at end-diastole

LVDs (mm): left ventricular dimension at end-systole

HR：heart rate

SV (mL): strole volume

CO (mL/min): cardiac output

IVSTd (mm): interventricular septum thickness at end-diastole

PWTd (mm): posterior wall thickness at end-diastole

3) The purpose of this study you wrote is twofold. On the other hand, only one is shown in the conclusion. Make sure that the purpose and the conclusion are matched as there is no discrepancy.

6. PLOS authors have the option to publish the peer review history of their article (what does this mean?). If published, this will include your full peer review and any attached files.

Reviewer #1: No

Reviewer #2: **Yes: **Michiaki Higashitani

---

## [Author Response · Author response to Decision Letter 0]

5 Apr 2022

Reviewer#1:

In this paper, the authors have focused on NOX2 and NOX4, which are expressed in cardiac myocytes among several NOX subunits. This choice makes sense. In particular, the authors focused on NOX4, which is highly responsive to LPS stimulation and provides an excellent research data and discussion of TLR4-NOX4 signaling in cardiac myocytes.

1) However, in cardiac myocytes, not only NOX4 but also NOX1/NADPH oxidase is strongly involved in endotoxin-induced cardiac myocytes apoptosis. Therefore, in the Discussion section, I recommend you to make some inferences about the relationship between NOX1 and NOX4.

Response:

We could not exclude the involvement of ROS derived from NOX1, a minor NOX subtype in the heart, in the development of PG-LPS-mediated cardiac dysfunction, because NOX1 might play a role in cardiac myocyte apoptosis after exposure to Escherichia coli LPS [1, 2]. Also, Helicobacter pylori LPS induces ROS through the activation of TLR4-NOX1 signaling, leading to apoptosis of guinea pig gastric mucosal cells and gastric ulcer formation [3]. These data might suggest that NOX1 is involved in the process of PG-LPS-mediated cardiac dysfunction. However, we think this is unlikely, because NOX1-mediated cardiac myocyte apoptosis is induced through decreased Akt phosphorylation in the heart of Escherichia coli LPS-treated mice [2], which is in contrast to the increased Akt phosphorylation in the heart of PG-LPS-treated mice (Fig 6A). 

We incorporated the above sentences in the discussion with new references (Page 34, Lines 6-16).

2) Similarly, please mention briefly here that no LPS-TLR4-NOX2 signaling was observed in cardiac myocytes as shown in Figure 4E. Since NOX2-derived ROS kills and degrades bacteria, there may be a reason why NOX2 is not responsive to LPS-TLR4 stimulation in cardiac myocytes.

Response:

ROS derived from TLR4-NOX2 signaling is known to be involved in killing and degrading microorganisms, contributing to host defense function in patients with infectious diseases [4, 5]. Therefore, we anticipated that NOX2 expression might be increased in the heart of PG-LPS-treated mice, but in fact we found NOX2 expression was similar among the four groups, as shown in Fig 4E. We do not know why NOX2 expression was not increased in the heart after PG-LPS infusion. PG was recently reported to circumvent the NOX2-mediated antimicrobial defense system, potentially enabling persistence in the tissue that supports the teeth [6], and so it is possible that PG-mediated circumvention of the host defense system via NOX2 might occur in the heart of PG-LPS treated mice. 

We incorporated the above sentences in the portion of discussion with new references (Page 32, Line 15-Page 33, Line 7).

P.S. In the description of Trichrome Statin Kit (#HT15-1KY; Sigma-Aldrich, St. Louis, MO); USA is required after MO in parenthesis.

Response:

Done as requested (Page 10, Line 7).

Reviewer#2:

Thank you for submitting to PLOS ONE.

I’m very interested in your article “Role of TLR4 signaling on Porphyromonas gingivalis LPS-induced cardiac dysfunction in mice”.

I recognize that the main points of this paper are the following two.

Immune system activation induced by Lipopolysaccharide on gram-negative bacterial pathogen known to be the cause of periodontitis, induce cardiac dysfunction and myocardial remodeling via the TLR4 signal. Furthermore, it was suggested that the TLR4-NOX4 pathway previously reported in the heart failure model may play an important role.

I sound that this paper is suitable for publication in PLOS ONE. 

However, one major revision and some minor revision are required.

Major revisions:

The target of your study is myocardial cell damage and cardiac dysfunction. However, cardiovascular events mediated by arteriosclerosis are discussed in introduction. Originally, if you would consider cardiovascular events related to atherosclerosis, coronary artery or vessels must be also investigated. I think introduction should focus on myocardial cell damage. For example, how about changing the reference 1) 12) to other paper such as heart failure model related to myocardial cell damage and discussing it?

Response:

Thank you for your suggestion. We replaced references #1 and #12 with new references and modified the text as shown below to focus the introduction on myocardial cell damage in mice after chronic PG-LPS infusion.

Page 5, Lines 4-5

It is also associated with an increase of myocardial cell damage and heart failure [7, 8].

Page 6, Line 15-Page 7, Line 2

More importantly, to our knowledge there has been no mechanistic study of the myocardial effects of persistent subclinical exposure to PG-LPS, even though subclinical endotoxemia was previously reported to induce myocardial cell damage and heart failure [9]..

In addition, we replaced reference #4 with a new reference and modified the following sentences.

Page 5, Lines 11-15

. But, even though PG-LPS is widely accepted as a contributor to PD-induced systemic inflammation, and patients with high anti-PG antibody levels were reported to exhibit a high rate of heart failure compared to the low antibody group [10], little is known about the precise relationship between PG-LPS and cardiac dysfunction. 

Minor revision:

1) You should write P-value at every item in all tables, even if there is no significant difference.

Response:

We incorporated the required data in S2 data in the revised manuscript. We also found miscalculations in the original Table 1, and have corrected them. The changes do not affect the results or conclusions of this manuscript.

2) The order and abbreviation of the items in the Table 2 A&B have to be changed as follows, 

Order

 1 EF

 2 LVEDV

 3 ESV → LVESV

 4 FS

 5 LVIDd → LVDd

 6 LVIDs → LVDs

 7 HR

 8 SV

 9 CO

 10 IVSDd

 11 LVPWTd → PWTd

Moreover, except LVSTs&LVPWT, and change Abbreviation details below.

WE(%): ejection fraction

LVEDV (mL): left ventricular end-diastolic volume

LVESV(mL): left ventricular end-systolic volume

FS(%): fractional shortening

LVDd (mm): left ventricular dimension at end-diastole

LVDs (mm): left ventricular dimension at end-systole

HR: heart rate

SV (mL): stroke volume

CO (mL/min): cardiac output

IVST (mm): interventricular septum thickness at end-systole

PWTd (mm): posterior wall thickness at end-diastole

Response:

Thank you. We have modified the table as you indicated.

3) The purpose of this study you wrote is twofold. On the other hand, only one is shown in the conclusion. Make sure that the purpose and the conclusion are matched as there is no discrepancy.

Response:

We modified the conclusion as follows (Page 37, Lines 2-7).

Immune system activation induced by chronic PG-LPS infusion contributes to the development of myocardial remodeling and cardiac dysfunction via activation of TLR4 signaling. Furthermore, activation of the TLR4-NOX4 pathway, which was previously reported to induce heart failure [11, 12], might play a role in PG-LPS-mediated cardiac dysfunction. Consequently, pharmacological inhibition of the TLR4-NOX4 pathway might be a promising strategy to decrease ROS production in patients with periodontitis.

References

1. Byrne JA, Grieve DJ, Bendall JK, Li JM, Gove C, Lambeth JD, et al. Contrasting roles of NADPH oxidase isoforms in pressure-overload versus angiotensin II-induced cardiac hypertrophy. Circ Res. 2003;93(9):802-805. https://doi: 10.1161/01.res.0000099504.30207.f5. PMID: 14551238.

2. Matsuno K, Iwata K, Matsumoto M, Katsuyama M, Cui W, Murata A, et al. NOX1/NADPH oxidase is involved in endotoxin-induced cardiomyocyte apoptosis. Free Radic Biol Med. 2012;53(9):1718-1728. https://doi:10.1016/j.freeradbiomed.2012.08.590. PMID: 22982050.

3. Kusumoto K, Kawahara T, Kuwano Y, Teshima-Kondo S, Morita K, Kishi K, et al. Ecabet sodium inhibits Helicobacter pylori lipopolysaccharide-induced activation of NADPH oxidase 1 or apoptosis of guinea pig gastric mucosal cells. Am J Physiol Gastrointest Liver Physiol. 2005;288(2):G300-G307. https://doi: 10.1152/ajpgi.00274.2004. PMID: 15458921.

4. Bedard K, Krause KH. The NOX family of ROS-generating NADPH oxidases: physiology and pathophysiology. Physiol Rev. 2007;87(1):245-313. https://doi: 10.1152/physrev.00044.2005. PMID: 17237347.

5. Lv J, He X, Wang H, Wang Z, Kelly GT, Wang X, et al. TLR4-NOX2 axis regulates the phagocytosis and killing of Mycobacterium tuberculosis by macrophages. BMC pulmonary medicine. 2017;17(1):194. https://doi: 10.1186/s12890-017-0517-0. PMID: 29233104.

6. Roberts JS, Atanasova KR, Lee J, Diamond G, Deguzman J, Hee Choi C, et al. Opportunistic pathogen Porphyromonas gingivalis modulates danger signal ATP-mediated antibacterial NOX2 pathways in primary epithelial cells. Front Cell Infect Microbiol. 2017;7:291. https://doi: 10.3389/fcimb.2017.00291. PMID: 28725637.

7. Chen TS, Battsengel S, Kuo CH, Pan LF, Lin YM, Yao CH, et al. Stem cells rescue cardiomyopathy induced by P. gingivalis-LPS via miR-181b. Front Cell Infect Microbiol. 2018;233(8):5869-76. https://doi:10.1002/jcp.26386. PMID: 29226955.

8. Matsuo I, Ohnuki Y, Suita K, Ishikawa M, Mototani Y, Ito A, et al. Effects of chronic Porphylomonas gingivalis lipopolysaccharide infusion on cardiac dysfunction in mice. J Oral Biosci. 2021;63(4):394-400. https://doi:10.1016/j.job.2021.10.001. PMID: 34757204.

9. Lew WY, Bayna E, Molle ED, Dalton ND, Lai NC, Bhargava V, et al. Recurrent exposure to subclinical lipopolysaccharide increases mortality and induces cardiac fibrosis in mice. PLoS One. 2013;8(4):e61057. https://doi: 10.1371/journal.pone.0061057. PMID: 23585870.

10. Aoyama N, Kure K, Minabe M, Izumi Y. Increased heart failure prevalence in patients with a high antibody level against periodontal pathogen. Int Heart J. 2019;60(5):1142-6. https://doi:10.1536/ihj.19-010. PubMed PMID: 31447467.

11. Kuroda J, Ago T, Matsushima S, Zhai P, Schneider MD, Sadoshima J. NADPH oxidase 4 (Nox4) is a major source of oxidative stress in the failing heart. Proc Natl Acad Sci USA. 2010;107(35):15565-15570. https://doi:10.1073/pnas.1002178107. PMID: 20713697.

12. Kuroda J, Sadoshima J. NADPH oxidase and cardiac failure. J Cardiovasc Transl Res. 2010;3(4):314-320. https://doi:10.1007/s12265-010-9184-8. PMID: 20559780.

---

## [Decision Letter · Decision Letter 1]

16 May 2022

Role of TLR4 signaling on Porphyromonas gingivalis LPS-induced cardiac dysfunction in mice

PONE-D-21-32097R1

Dear Dr. Okumura,

We’re pleased to inform you that your manuscript has been judged scientifically suitable for publication and will be formally accepted for publication once it meets all outstanding technical requirements.

Kind regards,

Tomohiko Ai, M.D., Ph.D.

Academic Editor

PLOS ONE

Additional Editor Comments (optional):

Reviewers' comments:

Reviewer's Responses to Questions

**Comments to the Author**

1. If the authors have adequately addressed your comments raised in a previous round of review and you feel that this manuscript is now acceptable for publication, you may indicate that here to bypass the “Comments to the Author” section, enter your conflict of interest statement in the “Confidential to Editor” section, and submit your "Accept" recommendation.

Reviewer #2: All comments have been addressed

2. Is the manuscript technically sound, and do the data support the conclusions?

Reviewer #2: Yes

3. Has the statistical analysis been performed appropriately and rigorously? 

Reviewer #2: Yes

4. Have the authors made all data underlying the findings in their manuscript fully available?

Reviewer #2: Yes

5. Is the manuscript presented in an intelligible fashion and written in standard English?

Reviewer #2: Yes

6. Review Comments to the Author

Reviewer #2: I have determined that your response meets the acceptance criteria and your paper is well worth publishing in PLOS ONE.

7. PLOS authors have the option to publish the peer review history of their article (what does this mean?). If published, this will include your full peer review and any attached files.

Reviewer #2: No

---

## [Editor Report · Acceptance letter]

23 May 2022

PONE-D-21-32097R1 

Role of TLR4 signaling on *Porphyromonas gingivalis* LPS-induced cardiac dysfunction in mice 

Dear Dr. Okumura:

I'm pleased to inform you that your manuscript has been deemed suitable for publication in PLOS ONE. Congratulations! Your manuscript is now with our production department. 

Kind regards, 

on behalf of

Dr. Tomohiko Ai 

Academic Editor

PLOS ONE